# Scope and Impact of Visualization in Training Professionals in Academic Medicine

Venkat Bandi*
Department of Computer Science
University of Saskatchewan

Debajyoti Mondal†
Department of Computer Science
University of Saskatchewan

Brent Thoma‡
Department of Emergency Medicine
University of Saskatchewan

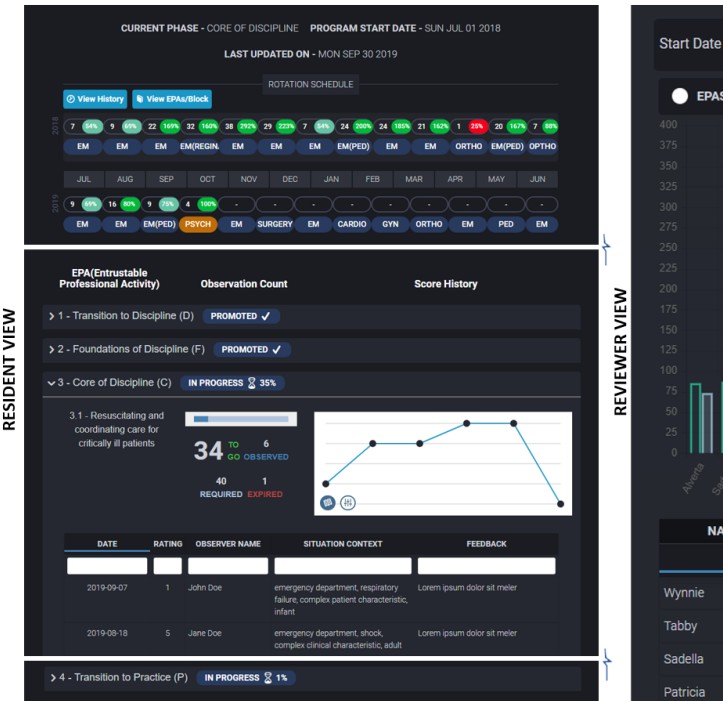
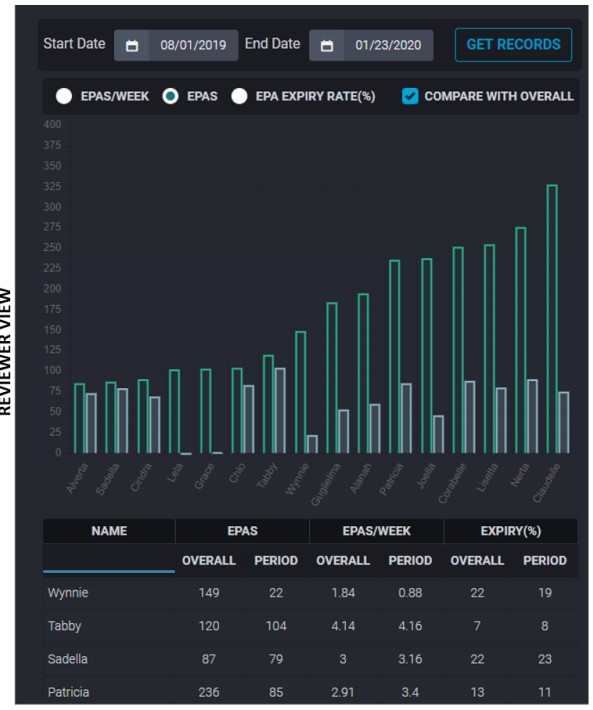

Figure 1: Different Views of our visual platform (on an anonymized dataset): (left) Resident view. (right) Reviewer view.

## ABSTRACT

Professional training often requires need-based scheduling and observation-based assessment. In this paper, we present a visualization platform for managing such training data in a medical education domain, where the learners are resident physicians and the educators are certified doctors. The system was developed through four focus groups with the residents and their educators over six major development iterations. We present how the professionals involved, nature of training, choice of the display devices, and the overall assessment process influenced the design of the visualizations. The final system was deployed as a web tool for the department of emergency medicine, and evaluated by both the residents and their educators in an uncontrolled longitudinal study. Our analysis of four months of user logs revealed interesting usage patterns consistent with real-life training events and showed an improvement in several key learning metrics when compared to historical values during the same study period. The users' feedback showed that both educators and residents found our system to be helpful in real-life decision making.

*e-mail: venkat.bandi@usask.ca
†e-mail: d.mondal@usask.ca
‡e-mail: brent.thoma@usask.ca

**Index Terms:** Human-centered computing—Visualization—Visualization toolkits; Applied computing—Learning management systems—Visualization design and evaluation methods

## 1 INTRODUCTION

Data visualization systems are increasingly being used to analyze educational data and learners' behavior in recent years [5,16,17,39,43]. Online learning management systems [9, 11, 18] often integrate educational dashboards that help learners to understand their performances, and educators to analyze learners' progress and needs. Learning analytics research that utilizes such dashboards [24, 30, 39] examines how such information representation could be used to motivate learners to improve their performances. However, such studies are usually based on educational theories and put less emphasis on cutting-edge techniques in visualization research [39]. Most innovative visualization dashboards focus on helping educators to understand learners' performances at a glance, and perform comparative analysis of the learners' progress [15]. Visual analytics research that helps examine the educational data coming from online course platforms, has started to bridge the gap between learning analytics and educational data visualization [5, 16, 17, 43].

Although much effort has been invested in understanding the scope of visualization in term-based courses that follow structured (weekly/biweekly) activity schedules over a fixed academic term [15, 22, 42], the usefulness or need of visualization in the context of managing professional training is less explored. The scenario is

particularly distinct in the following aspects:

- Professional training is often semi-structured and scheduled based on the need of an individual learner.

- A large part of the training may rely on observation-based assessment.

- Sometimes comparison among the learners is not meaningful due to the unique training characteristic for individual residents.

This motivated us to examine whether visualization systems can still be used to assist the learners and educators in such a context, and if so, then how the visualization views should be designed to help users perform their intended tasks.

In this paper we concentrate on a context of academic medicine, where qualified medical physicians (residents) practice their profession under the supervision of a senior clinician (educators/reviewers). All the aspects of professional training mentioned above are prevalent in a residency training program. To understand the scopes and needs of visualization, we collaborated with a group of residents and clinicians in the department of emergency medicine. We first identified a set of requirements from the perspective of the residents and their reviewers. We then followed an action design research framework [26] and created two interactive views through agile development [34]: *Resident View* (available for residents and reviewers to look at a resident's performance individually) and *Reviewer View* (available only for reviewers to compare multiple residents in the learning program at a glance).

The visualization system was designed over four focus group discussions, six major development iterations, and several agile scrum meetings during the development process [34]. We evaluated the final system through an uncontrolled longitudinal study over four months and through feedback sessions with the residents and reviewers. The results revealed that both the residents and reviewers found the system to be useful in assisting them with their learning and training activities. We further evaluated the change in several learning metrics after the introduction of our system compared to the previous academic year. The results revealed a marked improvement (e.g., 12% increase in residents' engagement and 22% increase in educators' feedback length) indicating that our visualization views were successful in improving the training and learning in an academic medicine environment.

Our contribution in this paper can be summarized as follows.

(1) Identification of the scope of visualization in meeting the needs of learners and educators in a medical training program.

(2) Design of a visualization system that helps residents to understand and track their performances and activities, and educators to make decisions in the training process.

(3) Evaluation of the system by 6 medical practitioners and 16 residents in an uncontrolled longitudinal study over four months, which shows the potential of the system in assisting users in the residency training process.

## 2 RELATED WORK

In this section, we review the literature on the use of visualization in learning analytics, visualization dashboards for understanding the users by analyzing data related to online course platforms, and educational data visualization in the context of medical training.

### 2.1 Visualization in Learning Analytics

Learning analytics are increasingly being adopted in large scale organizational systems, particularly in the field of higher education to help understand and optimize learning in varied environments [36]. Learning analytics research often relies on various forms of information representation to motivate learners, where those representations are inspired by educational and learning theories [41]. A dashboard can facilitate social comparison with the other learners by showing the learner's position among the peers, or can present a comparison with an earlier self by showing the learner's progress, or can show the progress towards achieving a goal [24, 30]. This line of research focuses on how various types of learners learn, and on the need to create personalized visualizations to motivate learners to improve their performances [29]. In our context, the residents have a diverse set of needs and goals, which makes social comparison harder compared to the other representations (self-comparison and progress towards the goal).

In a recent survey on visual learning analytics, Vieira et al. [39] observed that most learner-facing dashboards created for learning analytics, are limited to standard charts and plots even though there exist sophisticated visualizations (interactive multiple linked-view visualization [6]) and static dashboards [15] for presenting data that have been designed by visualization experts. A possible reason for this gap stems from the lack of collaboration between end-users in educational programs and visualization experts. Since traditional dashboards commonly focus on structured academic courses, they need to be adapted by visualization experts for specialized educational programs such as residency training, based on the need of the residents and their educators.

### 2.2 Visualization Dashboards

Dashboards are commonly described as a form of visual representation that helps monitor conditions, facilitate understanding, or motivate users to progress towards a goal [15, 40]. In general, dashboards may appear as a simple collection of chart types, which are interlinked through multiple coordinated views. However, a successful dashboard may have a profound impact on its users. The design process and technicalities behind a successful dashboard is challenging as it requires data-driven thinking, careful design of data access levels, and a good understanding of end-user flexibility and their visual literacy [33]. Some of the common uses of dashboards and visual encoding of data in general are to help decision-making, improve awareness and understanding about the context, facilitate exploration of the data, as well as infographics applications [10, 13, 14, 20, 27].

Dashboards often appear in online course systems. The popularity of massive open online courses (MOOC) has created many MOOC platforms and generated a vast amount of educational data. This has greatly advanced research in learning analytics that investigates learners' behaviors, learner-educator collaboration, and effective teaching methods. Most MOOC systems (e.g., Coursera [9], edX [11], FutureLearn [18]) integrate learner-facing dashboards to help learners understand and map their learning curve.

A rich body of research proposes various ways to use learning dashboards for motivating learners. A number of visual analytics systems have been proposed to analyze MOOC data to understand learners' behaviour [7, 39], video streams viewed by the learners [5, 43], communication among the learners on a forum [16, 17]. However, these approaches are not directly applicable to the professional training context, where the learners and educators frequently meet in face-to-face meetings for training and assessment activities.

### 2.3 Visualization in Medical Training

A number of commercial training management systems exist [8, 12] that focuses on general administrative processes such as session registration, scheduling, and reporting. But to the best of our knowledge, the scope of visualization for professional training is not yet well investigated, and a few studies have appeared only recently in learning analytics [28]. Recently, Boscardin et al. [3] have proposed some guidelines for designing a learning analytics dashboard in the medical education program, where they emphasized that visualization could play an important role in enhancing learners' experience.

Further, a survey on information visualization systems for exploring patient records concluded that while real medical data is often messy, incomplete and riddled with systematic errors, effective information visualization techniques can reduce this problem and provide valuable insights [32]. Since resident reviews are based on patient medical records, similar visualization techniques can be adopted to evaluate their performance from this data. This was demonstrated by Basole et al. [1] in a visual analytics system that looked at patient records to understand care process variation and presented a provider level performance summary among other results. This form of aggregated data visualization from alternative sources other than structured exams can help educators in understanding a medical learners' performance. In our context, we need to look at not only how medical learners perform in varied patient situations, but also combine this with their current learning trajectory to provide a summary visualization of their performance over time.

Residents also need heavy faculty support to follow the learning goals and develop skills [25]. Intuitively, a learner facing dashboard for residency training should help analyze how the residents self-reflect, interpret and act on the educators' feedback. However, determining how or whether the residents use the performance dashboard to guide learning varies widely among the students [21].

## 3 Application Background

Our investigation in this paper spans a professional training context in academic medicine. Our goal was to explore and understand the scope of visualization to improve the training and learning process in a residency program (a training program for medical physicians). The investigation was carried out in a collaboration with a group of 16 residents and 6 clinicians of the department of emergency medicine. The subsequent four sections describe the necessary background for problem formulation (Section 3.1), task abstraction (Section 3.2-3.3), and dataset characterization (Section 3.4).

### 3.1 Relevant Terminologies

**Resident** - A medical physician undergoing training in a specialized field in a supervised setting.

**Competence Committee** - A supervising committee that monitors a resident's progress and takes decisions regarding promoting a resident to the next phase of their learning.

**EPA** (Entrustable Professional Activity) - A task or a responsibility that a resident can be entrusted to perform in a heath-care context to demonstrate competence in that activity (e.g., EPA 2.1-Initiating and assisting in resuscitation of critically ill patients) [38].

**EPA count** - This is the number of observations that have been made for a particular EPA type.

**Rotation Schedule** - A year-long schedule spanning 13 blocks (rotations) with 4 weeks each, in which a resident works across different departments (Cardiology, Trauma, Neurology, etc.) to gain diverse clinical exposure.

**Training Phase** - A part of a resident's learning program. Residents are progressively promoted based on their performance through the following 4 phases before they finally graduate: Transition to Discipline, Foundations of Discipline, Core of Discipline, and Transition to Practice.

### 3.2 Problem Domain

The residents are recorded throughout their learning phase by a faculty or a senior resident in that medical program. The residents' scores depend on how they respond to patients on the tasks assigned to them. To ensure that residents have a broad understanding of their field, they are required to attend to a wide variety of medical scenarios. In each of these medical scenarios, the residents are rated on a 5 point scale and provided with feedback on how to improve.

Although this information is collected on a case by case basis, it can also be collated across multiple levels from various sources to gain a wealth of information about not just individual residents, but the reviewers and the entire training program.

### 3.3 Requirement Analysis

To develop our visualization platform, we first conducted focus group studies with the intended users (residents and users). We then extracted a list of analytical tasks.

#### 3.3.1 Focus Groups

To analyze the requirements of the visual analytic system, we conducted four focus group meetings, three with residents and one with competence committee members. Each meeting took approximately one hour.

*Focus group with the competence committee:* In the meeting with the competence committee, the participants were asked on what information they would like to know about a resident to review them and how they would want it to be presented.

*Focus group with the residents:* We used the first two of the three meetings with the residents for requirement gathering, and the third meeting to refine our design choices.

The first focus group meeting with residents involved 10 residents from the emergency medicine program. They were asked to describe all the data they wanted to highlight to their reviewers, and also to see for themselves for their personal reflection. They were then asked to make rough sketches on how they wanted this information to be visualized. We followed this focus group meeting with a second meeting two months later with 5 residents from the same program. The meeting was conducted as a think-aloud session, where participants were shown an initial list of requirements and a basic sketch of how the data would be presented and asked to comment on how they would improve it. A third focus group meeting was conducted four months from the second focus group meeting. This meeting was held with the residents after our third major design iteration to refine our original requirements and get feedback on the system prototype.

#### 3.3.2 Decision-Making Tasks

We collated all the requirements into two themes, each to be presented as a module in our visualization platform (Fig. 1). We observed that the requirements for the residents are also important to the reviewers to assess residents. Hence our design allowed reviewers to access the resident view.

**Resident View**

$Q_1$. Where is every resident currently in their training program, where were they before and where will they be next in a given academic year?

$Q_2$. Is a resident meeting the required number of EPA observations needed in a specific rotation (Cardiology,Trauma etc.) as per the resident's schedule?

$Q_3$. How is a resident performing over time for a specific EPA? Does a resident meet the minimum number of EPA observations required (for a specific EPA) to demonstrate competence as per standard requirements?

$Q_4$. In a given EPA, has the resident achieved the required diversity in terms of patient demographics (child, adult and senior cases) and medical scenarios?

**Reviewer View**

$Q_5$. How is a resident performing compared to all the other residents overall and in a given time period?

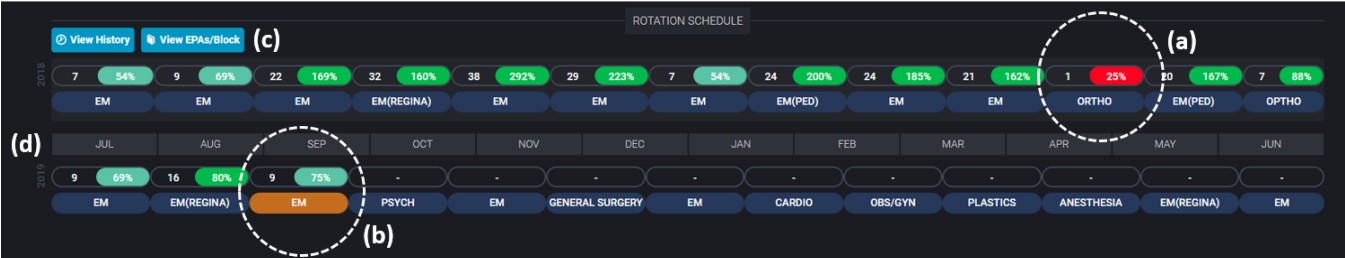

Figure 2: Rotation schedule of a single resident with the following components: **a)** Rotation with very low EPA count visually encoded in red to stand out. **b)** Current rotation with its corresponding rotation above showing that the resident has 9 EPA observations currently with a target achievement rate of 75%. **c)** *View History* and *View EPAs/Block* toggles which display historical rotation schedules and EPA count information layers. **d)** Reference timeline showing the academic year and dividing the current and historical years.

## 3.4 Data Characterization

To answer the questions $Q_1$-$Q_5$, we collected and combined two datasets: One is the residents' EPA assessment data and the other is their rotation schedule.

### 3.4.1 EPA Assessment Data

Residents are monitored every time they perform an EPA observation by a faculty or a senior resident. They receive a rating on a 5 point scale, as well as qualitative feedback. This data is then entered into Mainport, an online website maintained across Canada by the Royal College of Physicians and Surgeons. For the purpose of our visual platform, we exported this data from the Mainport website and uploaded it into our system on a weekly basis. Since the start of this project, we have collected around 3500 EPA assessment records for 16 residents. Every assessment record has the following information: *Resident Name, Reviewer Name, EPA Type, Rating, Feedback, Observation Date, and Situation Context.*

*Resident Name* and *Reviewer Name* identify the residents and reviewers, respectively.

*Observation Date* is the day on which the EPA was completed.

*EPA Type* is a numerical label with a description of the professional activity. Every EPA assessment includes information about the patient demographics (e.g. age, gender) and the type of clinical presentation (e.g., shock, cardio-respiratory arrest, sepsis, etc.), which is provided as a list of values in *Situation Context.*

*Feedback* comprises a description of the actual situation and how the resident responded to it with additional comments on how to improve in the future.

*Rating* is an ordinal value provided using an O-Score Entrustability Scale [19] with **1** being the worst and **5** being the best in the following order: I had to do (**1**), I had to talk them through (**2**), I had to prompt them from time to time (**3**), I needed to be in the room just in case (**4**), and I did not need to be there (**5**).

### 3.4.2 Schedule Information

Every academic year, residents go through a 13 block rotation schedule. This schedule is unique for every resident and depends on their learning plan. Along with this information, we also record important dates such as when the resident started the residency program and the different dates on which the resident was promoted to the various training phases. In a data pre-processing stage, we combined the schedule and the training phase promotion dates with the EPA assessment data to tag every record with two additional identifiers: *Rotation Tag* and *Phase Tag*. They are both calculated by placing the *Observation date* on a time scale to identify which rotation and training phase residents were in when they completed an EPA.

## 4 VISUALIZATION DESIGN

In this section, we describe every part of our system and the visual encoding used in each to answer the questions of various user groups (residents, reviewers, and program evaluator).

### 4.1 Resident View

The resident view focuses on $Q_1$-$Q_4$ (discussed in Section 2.2) for an individual resident, i.e., the rotation schedule of the resident, acquirement of various EPA types, EPA-specific performance trend, and a quick review of the resident's recent activities.

#### 4.1.1 Rotation Schedule

A Gantt chart is a common type of chart used to illustrate a project schedule, but this takes significant vertical space (one row per project activity). However, in a resident's rotation schedule, no pair of rotations (e.g., Cardiology and Trauma) overlap. Hence we created an interval representation for the activity blocks, where the length of an interval represents the length of the rotation and collapsed our design vertically, as illustrated in Figure 2.

The interval chart is a combination of chronological rotational information along with EPA completion rate. To answer question $Q_1$ (i.e., the resident's current position in the training schedule), we first create a time axis consisting of a series of rotations with month names on them. The rotation schedule starts from July and ends in June, which represents the academic year of the medical program. We then use the scale itself as a horizontal separator between the current academic year and the historical years (Figure 2(d)). Specifically, we show the rotations of the current academic year below the time axis, whereas the past years are depicted above.

For every academic year, we create a series of 13 consecutive rotations with the name of the rotation inside the block and arrange them chronologically based on the start date of each block. We highlight the current block the resident is in with an orange shade (Figure 2(b)) so that users can easily identify it. This provides an overview of what rotations the resident have completed, where the resident is now, and what rotations the resident will be completing next. Such an overview is not only important for the resident to self-reflect, but also to the reviewers to understand the resident's context. The uniqueness of every resident's schedule further justifies the need for such a concise representation.

#### 4.1.2 EPA Count in a Rotation

To answer question $Q_2$ on whether a resident is achieving the required EPA count in a given rotation, a second block is provided right above every rotation block consisting of 2 numbers. The first is a number aligned to the left providing the sum of the EPA counts for all EPA types that a resident performed in that rotation. The second is the EPA target achievement, which is defined as the percentage of EPA observations completed by a resident to the target number of

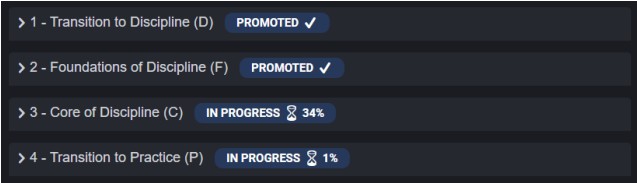

Figure 3: EPA Overview Table for a single resident currently in the training phase Core of Discipline.

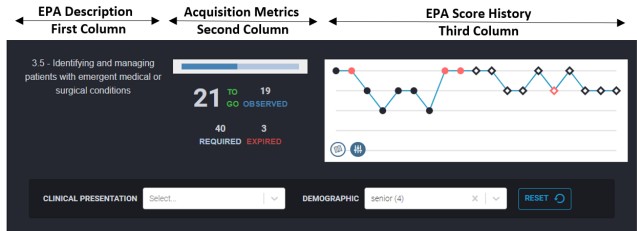

Figure 4: An example EPA 3.5 in phase **Core of Discipline** with both date filter (diamond points) and demographic filter active (red points).

EPA observations required in a given rotation. The target is set by the competence committee based on the historical performance of residents.

While looking at the target achievement rate, competence committee members are interested in identifying under-performers more than over-achievers, because in rotations with a small target count, residents can often over perform, leading to very high percentages. We adopted a capped color scale instead of positional encoding because theoretically, the maximum target achievement rate is unbounded. We filled the target achievement number inside a cylindrical bubble and color the bubble using a linear scale that ranges from dark-red to dark-green for the percentage value of 0% to 100%. The scale is capped at 100% (i.e., any value above 100% remains dark-green). This ensures that rotations, in which a resident under performs, can be easily identified, but the over-performing rotations will not stand out. Even though the target achievement rate is visually encoded by a color scale, we still explicitly list the numbers to mitigate any effect which could be associated with our choice to clamp the color scale on the higher end. Thus while under performing blocks visually stand out, the information about other blocks can be gleaned by reading the achievement rate text on every block.

Two buttons to toggle the visibility of the historical years and EPA target achievement rate are provided, and are turned off by default. This design ensures visual clarity by showing only basic information and hiding additional information under a collapsed view, which can always be brought up on demand. This option to toggle the visibility of visual blocks and charts has been reused in several other components in our visualization platform and combined with the option to filter records based on a time interval follows the visual information-seeking mantra of "Overview first, zoom and filter, then details on demand" [35].

### 4.1.3    EPA Overview Table

The entire list of EPA types that residents are required to complete are broken down into four groups based on the training phase during which a resident is supposed to complete them and are numbered accordingly.

The serial ordering of the EPAs provides a natural choice for presenting them in a tabular format. We provide 4 blocks arranged vertically with the oldest starting from the top for every training phase as shown in Figure 3. For every phase, we present a comple-

tion rate which is calculated as the ratio of the sum of acquired EPA observations to the sum of required EPA observations for every EPA in that phase.

Although residents generally complete the EPAs of their current training phase before they pick up EPAs of later phases, there are exceptions. Due to various external factors such as their rotation schedules and the nature of medical cases of the patients they attend to, residents can occasionally end up completing EPAs which are not in their current training phase. This means residents can have a non-zero completion rate for training phases that they have not yet started.

To present this information, we use a combination of icons, numbers and textual labels, as shown in Figure 3. The icons are used for quick lookup on which phases are complete and which are in progress. The labels are used to annotate the icons for the users unfamiliar with the meaning of the icons. The numbers, which represent the completion rate, are not shown for phases that are already complete (since they are not important to the users). Every row in the overview table acts as an accordion that can be opened and collapsed (Figure 1(left)), and the encoding of content in the accordion is discussed further in the following section.

### 4.1.4    Individual EPA Overview

When a user clicks on a row in the overview table, a list of all EPA types in that training phase is shown in a tabular layout with three columns, as shown in Figure 4.

*First Column:* In a particular training phase, the first column provides the identifying serial number for all the EPAs in this phase, as well as textual information describing the corresponding medical scenarios.

*Second Column:* In a particular training phase, the second column depicts the resident's position in each EPA of this phase. Each row represents a single medical scenario, and the second column of the row consists of two items (Figure 4). The first is a bullet chart [15] that visualizes whether a resident has met the target number required for that EPA. We visually encode the percentage of completion for a given EPA as the length of the bar inside the bullet chart. We do not use any scale on the chart, as it is there to assist users in gaining an approximate idea of the percentage just by a quick lookup. We also cap the bullet chart at 100% so if a resident over performs in an EPA it does not stand out. A common visualization strategy is to stack the excess performance using the idea of a Horizon chart [23], but we did not use this due to the unfamiliarity of such a chart among the users. However, this information can be inferred (if needed) from a section of numbers beneath the chart.

A set of four numbers below the bullet chart provide additional information (Figure 4) such as the number of observations that are still needed to complete the EPA, the number of observations completed so far, the overall number of observations required, and the total number of expired observations (i.e., the EPA observations that were not completed by reviewers within a month of the date of entry). Since the number of EPA observations that still need to be completed is an important metric for a resident, we present this information (with the text 'X to go') at the beginning of the other numbers in the series. In addition, we emphasize it by choosing a slightly bigger font-size than the rest.

The numbers are associated with the labels 'to go', 'completed', 'required', and 'expired', respectively. We choose the color based on what the label represents. For example, the 'expired' label is coded in red to draw attention of the user because every time an EPA expires, it is an EPA that even though a resident has completed was never reported. This is to emphasize that the residents should track their EPAs and make sure that they are completed on time. The label 'completed' is colored in blue, to match the color of the bar in the bullet chart as they both represent the same information. The labels 'required' and 'to go' are colored in a lighter shade of blue and green

respectively to bring a color balance between the four numbers.

*Third Column:* In a particular training phase, the third column visually depicts a resident's progress for all the EPA observations in that training phase. Columns 2 and 3 together answer $Q_3$ (i.e., the position and performance of a resident for a specific EPA). We opt for a simplistic line chart design with minimal clutter. Every record is represented as a point with the oldest record starting on the left. The points are arranged vertically using the O-Score Entrustability scale with 5 being the highest (resident managed the situation independently) and 1 being the lowest (senior doctor had to completely take over the situation). The better a resident performs in an EPA, the higher is the vertical position of the point in the chart. We use background lines to show the 5 levels, instead of labelling the points, to reduce visual clutter as the levels are easy to understand without providing additional context. An earlier iteration of the system did include Y axis labeling for each of the levels but several users remarked that this was unnecessary and fairly self evident and seemed particularly repetitive when several charts were visible.

A common design choice for representing a time series data is to use a line chart with a date/time axis, where the axis is scrollable or zoomable. However, for the x-axis, instead of using an exact date/time scale, we use a set of equally spaced points in chronological order, with the oldest being on the left of the chart. The reason for not using the observation dates is that the observation dates are irregular, not important for the users, and furthermore, the observations often span a large period of time, i.e., residents often complete EPA observations in chunks with long periods of time gaps in between.

Every point (observation score) can be hovered over with a mouse to bring up additional information about that record in the form of a pop-up. This includes observation date, feedback, the patient demographics and medical context of the case.

Finally, to answer $Q_4$ (i.e., diversity of the demographics covered by the resident), we provide two small buttons at the bottom-left corner of the chart. The first one can be clicked to see all the records in a table that can be sorted and filtered by columns. The second button brings up two filter lists that can be used to visually identify a particular record based on patient demographics or medical context. For example, if a user wanted to see which of the records were for senior patients, they could select the 'Senior' option from the drop-down list and the corresponding points (observation scores) would turn light red. We used a categorical color palette for the demographics.

Because every EPA observation record has an observation date associated with it, we can filter the records to look at a resident's performance in a given time period. However, this information is only useful when looked at in the context of the other records the resident has completed because every EPA observation is a test demonstrating competence and this can only be acquired over time. Hence instead of showing only those scores on the line chart, we change the shape of the observation points that lie within the selected date range. Regular records are shown as circular disks, while records that fall in the selected date period are shown as hollow diamond points (Figure 4). Several alternative design choices were also explored such as a single vertical line to distinguish all records that fall on one side of the line to be after a given date. However, when looking at specific time period with a definite start and an end date, two vertical lines would need to be used. This design could be further adapted to be a single partially transparent rectangle spanning the records in that time period to distinguish them. However, of all the possible design choices, users found the combination of the two visual encodings (shape and color) to be the most intuitive and so that design was ultimately adopted. An example of this can be seen in Figure 4 where we examine all the Senior cases in EPA 3.5 that fall in a particular time period, using a combination of the date filter and the demographic filter.

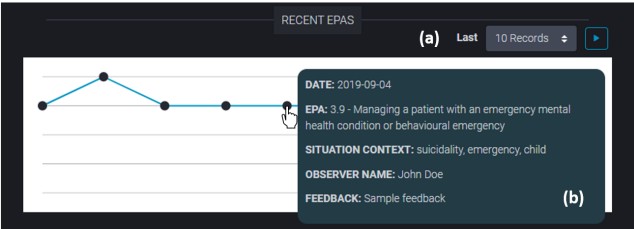

Figure 5: Recent History Chart. **a)** Filter to select the number of points to show in terms of last X months or last Y records. **b)** Popup screen displaying additional information about the point that is being hovered on by a mouse cursor (anonymized data).

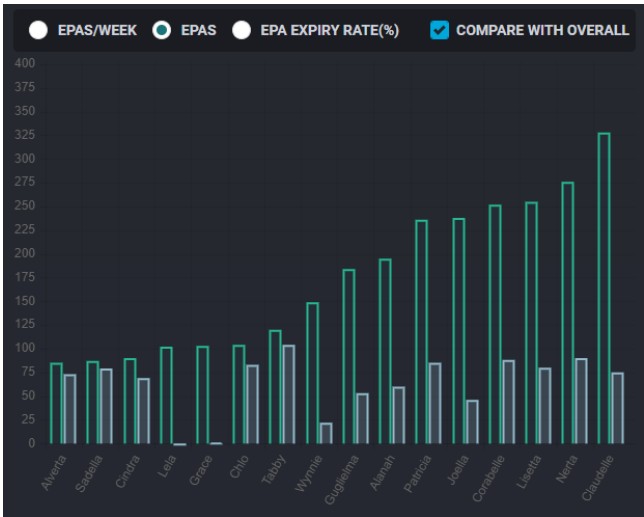

Figure 6: Review View comparing all residents with the green bars representing overall EPAs and the blue bars representing EPAs in a particular time period (anonymized data).

#### 4.1.5 Recent History Chart

This chart is meant to quickly lookup a residents' recent performance with the option to view records in the following ranges: last 10 days, last 25 days, last month and last 3 months. The chart does not visually distinguish the different EPA types (i.e., EPA-2.1 vs EPA-3.2), instead, it provides this and other additional information in a pop-up menu that can be invoked by hovering over a point, as shown in Figure 5.

### 4.2 Reviewer View

While assessing a resident in a residency program, a reviewer is often interested to see an overview of the resident's performance. Therefore, we made the resident view accessible to the reviewers. We were also interested to know whether comparing residents with their peers would be valuable, and we found mixed opinions. Some reviewers mentioned that they usually do not compare a resident's performance with peers as residents are assessed based on their unique rotation schedules and future plans. This is particularly different from a structured course context, where the learners are often assessed altogether by the same assignment or test. However, the majority of the reviewers still wanted to look at a high-level overview of how a resident is performing compared to the other residents (i.e., $Q_5$) in the same training phase, as it would give them an insight about the diversity in the residents' performances, and also help them in identifying outliers in the learning program.

In the comparative chart that we developed for the reviewers, we

present three metrics for each resident: *Total EPAs, Total Expired EPAs,* and *EPAs/Week*. The *Total EPAs* and the *Total Expired EPAs* are the count of all EPA observations and the count of all expired EPA observations across all EPA types, respectively. The metric *EPAs/Week* is an indication of a resident's weekly EPA observation acquisition rate, and is defined as the average number of EPA observations a resident acquires in a week across all EPA types. We chose a week as the time interval because during our pre-analysis of the data, residents were found to acquire EPA observations in small groups with multiple EPA observations in a single day followed by several days with no observations.

Since the data is categorical in nature, a bar chart is used for visual representation and users are provided with the option to select any one of the above-mentioned metrics using radio options in the panel above the graph, as in Figure 6. Users can look at the data in a particular time period, and also compare the value of a metric in a time period against the entire time a resident was active in the program. For example, in Figure 6, the green bars represent the Overall EPAs acquired by each of the residents over their entire program so far, and the blue bars represents the EPAs that the residents acquired in a particular time period that a user can set in a date filter (not shown in figure) above the chart. The checkbox *Compare with Overall* can be checked on or off to toggle the visibility of the green bars showing the overall values.

## 5 WEB IMPLEMENTATION

We developed our visual platform as a single-page web application that is linked to a NodeJS REST API in the backend, which in turn connects to a NoSQL MongoDB database, where the data is securely stored. We adopted a distributed architecture by using a REST API instead of the more traditional dynamic approach with pages being rendered on the server to ensure loose coupling, and easier scalability. The single-page application was built using a combination of *React.JS* [31] (web library used at Facebook for handling rapid data updates to the screen) and *D3.JS* [4] (web visualization library). Users access our system using a wide range of devices, i.e., smartphones, tablets and desktop computers. To ensure that the user experience was uniform across various screen sizes and orientations, we utilized a responsive design [37] as shown in Figure 7. The idea of using a collapsible overview table for visualizing blocks of information is very useful in devices with small screens, as the cells can be collapsed and expanded as needed. All visualizations in the system adapt based on the width of the screen, and are rendered in a scale and transform invariant SVG (Scalable Vector Graphics) format. Our design was implemented with the latest web standards and is available on GitHub[1] under an MIT license, which will allow other medical programs following a similar competency based learning approach to adopt the same system.

## 6 USER EVALUATION AND FEEDBACK

We deployed a stable version of the visualization system online for the department of emergency medicine at the university, and provided access to both user groups: Residents (16) and Reviewers (6). The users were allowed to utilize system for two weeks to familiarize them with the visualization platform. We then conducted an uncontrolled longitudinal case study by recording user logs for a period of 4 months on the website. We analyzed these logs to understand whether the users interact with the system, and if so, how various real-life scenarios influence the system use. We were interested in examining the change in the usage pattern as an indicator of whether or not the users found the system useful for their real-life decision making.

We also compared the EPAs acquired by residents during this study period with EPAs acquired by them the previous year in the

---

[1]github.com/kiranbandi/cbd-dashboard-ui

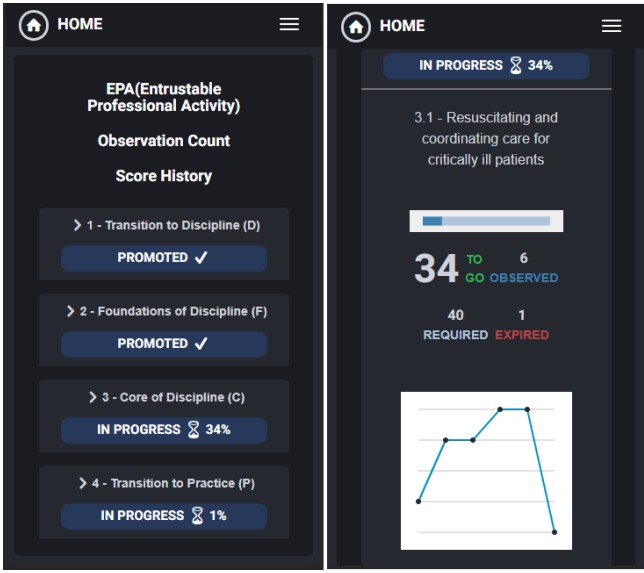

Figure 7: Screenshots of the system on a mobile screen showing responsive design.

same period during which they did not have access to our system. This was done to quantify the change in the different learning metrics brought about by the introduction of our system.

Finally, feedback was collected from the different user groups. We conducted feedback sessions with seven residents after they had considerable exposure to the system, and also collected feedback from five reviewers through individual meetings.

### 6.1 Analysis of User Logs

To quantify user engagement, we logged every user visit to the website. The logs included the name of the user, the time of visit, the user type and the page visited on the system. We recorded the logs for a period of 4 months. We selected the time period to intersect with two quarterly competency committee meeting dates for emergency medicine (September 10th and December 11th) as it would help us understand how the usage changes during and around these meetings.

We only kept the logs for the intended user groups, i.e, filtered out the logs of the developers, who were involved in the maintenance of the system, and administrators who periodically uploaded resident data onto the website.

Based on the user logs, we found that the resident view was the most visited part of our system. This is because of the larger number of residents, and also that it can be accessed by multiple user groups (residents to view their own data and reviewers to view data of all residents individually). We thus focus our analysis on the usage of the resident view.

#### 6.1.1 Usage Pattern Influenced by Assessment Meetings

To understand how user engagement changes over time we binned our logs into one week long intervals. We then grouped all the logs in each weekly interval based on the user type (resident or reviewer) and counted every log as one visit. The total number of weekly visits made by all users to the resident view reveals that the count peaked during Week 3 (September 8 – 14) and Week 16 (Dec 8 – 14) with a smaller peak during Week 9 (October 20 – 26), as shown in Figure 8. This could possibly be because the competence meetings happened on September 10th and Dec 11th. The smaller peak may have been due to a small internal review meeting that happened on Oct 23rd. This pattern shows the organic engagement that different

user groups have with our system and is also inline with comments made by residents during the feedback discussion such as *"I have looked at it prior to usually like my competency committee meetings. That is kinda when I mainly use it"*(P4).

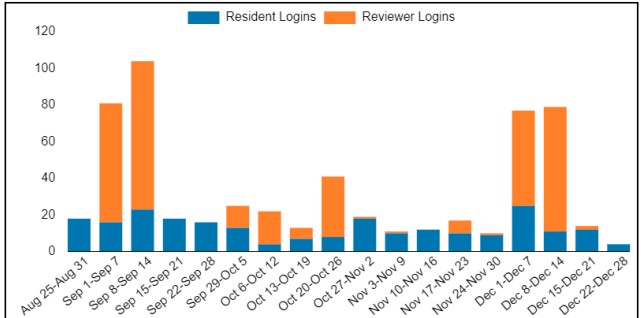

Figure 8: Number of visits made by different user groups to the Resident View during the Study Period.

The break-down of the visits based on the user types reveals that resident visits remained fairly consistent with a slight increase during the meeting week, but the number of visits made by reviewers changed drastically. As shown in the stacked bar chart in Figure 8, visits made by the reviewers were very high during the meeting week and the week before it and then dropped again after the meeting period.

### 6.1.2 Usage Pattern Influenced by Information Updates

The residents' weekly EPA scores are imported into our platform every Tuesday. Although the residents get their EPA scores and feedback over their email earlier than Tuesday, several residents mentioned that they log in every week especially on Wednesdays to check their progress, which is also an indication that they found the system useful to track their performances. We examined this by analyzing usage logs based on the days of a week and found a peak on Wednesdays, as shown in Figure 9(right). This clearly demonstrates the level of engagement some of the residents have with the visualization system, and how it has become an integral part of their learning program.

### 6.1.3 Usage Frequency vs. Expired EPAs

While analyzing the log, we observed two types of residents (frequent and non-frequent) users. We examined whether there is any difference between these two groups in any of the EPA related metrics.

We classified the residents into two above mentioned groups as follows: residents who visited the website at least once a week during the study period, and residents who did not. We then looked at different metrics of EPA acquisitions for the two groups and found that the rate of expired EPA observations for residents who visited the dashboard at least once a week was 3 times lower than the expiry rate for residents who did not visit as frequently. A potential reason could be that residents who frequently looked at the dashboard, ensured that all EPA observations they completed were submitted on time. However, we did not notice any interesting pattern in other metrics such as the number of EPA observations acquired and average EPA score.

### 6.2 Change in Metrics after System Adoption

To quantify if our system brought an improvement in the learning program of emergency medicine we identified four key metrics, which are some of the common measures monitored in the academic medicine program, and compared their change after the system was introduced. For this analysis, we looked at all the EPAs acquired by

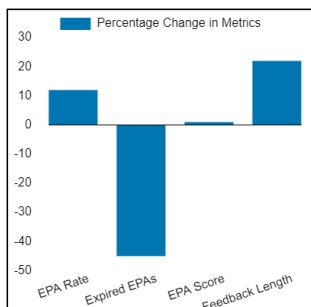
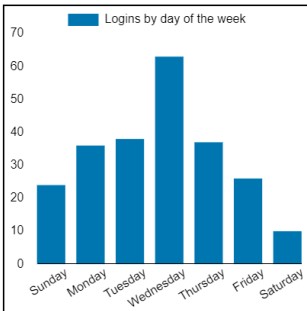

Figure 9: (left) Change in metrics after the introduction of our system. (right) Visits made by residents to the system distributed over days.

residents during the study period of four months after the system was introduced and compared them with the EPAs acquired by the same set of residents in the previous year during the same four months period. Before the introduction of our system the learners and educators were relying on a document based system that generated textual report. Our analysis revealed that there was positive improvement in three key metrics (EPA acquisition rate, Expired EPA count and Feedback Length) out of the four as shown in Figure 9 (left).

### 6.2.1 EPA Acquisition Rate

The first metric we looked at was EPA acquisition rate, we calculated this by averaging the total number of EPAs a resident acquires during each rotation normalized to the required count of EPAs in that rotation. This would reduce the bias that would be introduced when looking at residents in different rotations. For example, a resident in a Paediatric Rotation would require 10 EPAs per rotation while a resident in a Surgery Rotation might only need 8 EPAs per rotation. The EPA count normalized based on rotation would give us a metric that can indicate whether residents are acquiring the given number of EPAs in each rotation. Comparing this EPA rate during the study period revealed that there was a 12% improvement after the introduction of the system. A potential reason could be that our visualization system helped residents to gain a better understanding of their EPAs and hence, they could acquire the required targets in each rotation. Some for example commented that the system made them more organized and helped them plan better - *"I've made a plan for how many EPAs I need to get done per shift in the coming year, with a pretty good idea of which ones I'm more lacking in than others."*(P7)

### 6.2.2 Expired EPA Count

EPAs can often expire when supervising doctors do not submit them in time and, the onus of the responsibility falls on residents to keep in touch with the supervising doctor to ensure that their EPAs are submitted on time. With the adoption of our system expired EPA count fell by a drastic 45%. A possible reason for this could be residents having access to all their records on file in an easily accessible interface that lets them view their records sorted by time. This is beneficial to both the residents and the learning program as expired EPAs demonstrate wasted effort on behalf of both the resident and the supervising doctor.

### 6.2.3 Feedback Length

Qualitative feedback given during an EPA assessment can be crucial for residents as it can help them in understanding how they could have performed better in the given medical situation. Having access to a centralized system where they can look back on earlier feedback given to track their improvement meant that residents were more

proactive in asking for feedback during their EPAs. This could possibly be the reason for a 22% increase in the amount of feedback given during the study period. While word count alone is not indicative of feedback quality, there is medical education research which suggests that longer comments are predictive of higher quality feedback [2].

### 6.2.4 EPA Score

The average EPA score after the introduction of the system only improved by a very small percentage. This could possibly be because of the general learning nature of an EPA assessment where residents often acquire lower EPA scores when they perform in a particular medical situation the first time and then improve over time with experience. Thus for every EPA the score would ideally improve over time but this effect when averaged over several different EPAs among residents in different learning stages could become scrambled. We believe that in future research (once our system collects enough data over several years), EPA scores could instead be assessed independently for each EPA among residents in the same learning cohort.

## 6.3 Users' Feedback

### 6.3.1 Residents' Feedback

Residents can be broadly broken into two groups: those who visit the website only during review meetings, and others who use it on a frequent (weekly) basis. The difference between the groups can be partially explained by how well versed they are with the different aspects of the system.

Some residents seem to use the system extensively - *"I use it just as an interface just to get a better kind of gestalt view of how I'm doing. I can track my progress and EPA numbers per week which is important to me...The other thing I like to use it for is using it to kind of plan my life out because it has all of the dates of my rotations kind of in a convenient space"*(P2). We believe that as residents gain more experience in using the different components of the system, they may transform into frequent users of the system. Some residents mentioned that they use the system for casually looking at their EPAs - *"...I would go onto it just to peek at which EPA, to look at like how they've been trending and how I've been doing with most of them. But me personally, I haven't used it a ton. But that's probably 'cause I don't know all of the aspects of it that I can utilize"*(P1).

We have also seen that the residents found their own ways of using the system. For example, they sometimes used the visualization *"to identify gaps ... Like needing more assessments for a certain EPA"*(P6), or to *"plan for how many EPAs I need to get done per shift in the coming year, with a pretty good idea of which ones I'm more lacking"*(P7), or to use *"as a documentation of the conversation happening"*(P6).

### 6.3.2 Reviewers Feedback

We demonstrated the system to the reviewers and program evaluators in individual meetings. The reviewers appreciated the visual platform mentioning that the *"layout is good"*, *"it's way more user-friendly"* than the existing system, and also that they are getting used to the visualization platform over time: *"every time it's getting easier so that's a good thing"*. Reviewers also liked the simplicity of the design, e.g., *"just keeping it simpler and cleaner is better in my mind"*.

The visualization platform greatly helped the competence committee meeting: *"especially this last meeting was the best one I thought that we've had overall. It was the easiest meeting to get the data and put everything together"*.

The reviewers appreciated various visualization components. For example, they mentioned that the rotation schedule view (Figure 2) was *"very helpful just to actually visualize the number of EPAs they've had per block"*. A common use for the EPA overview

(Figure 4) was to *"look at all the outliers, if any of the EPA ratings that are 3 or less... then go through the actual feedback with those just to see if they're actually correlated"*. Both the demographics filter and the recent history chart (Figure 5) were helpful for the reviewers in their review. Some reviewers mentioned that they are able to perform even deeper investigations, e.g., *"to see if the numbers are low, why they're low, what rotation they were missing... if a rotation didn't go well... so-and-so was struggling... then you could see that their numbers were also low, then it would be a flag to talk to the resident to see what's going on"*.

On a question about whether they compare resident performances (Figure 6), some reviewers mentioned that they prefer *"comparing resident to a standard"* rather than peers because *"everybody's different"*. But some reviewers mentioned that *"I'll kinda compare them. Just to see if anybody's kinda really an outlier, falling behind"*. Sometimes they mentioned this to be useful for looking up *"how many EPAs they should be getting in that four-month timeline. So I compare them to their cohort"*.

The enthusiasm of the reviewers was eminent from their various expectations to further enrich our visualization system. For example, to have *"a summary since the last competency committee meetings"*, to integrate other datasets such as *"self-reflection of residents"*, and even to extend the system for the evaluation of the reviewers' feedback.

## 7 DESIGN CHOICES AND INSIGHTS GAINED

One of the most important insights that we gained in our study is how differences in professional training demand a design that is different from a dashboard developed for a typical structured course. The key design issue is that every resident is assessed based on their unique rotation schedule and need, whereas in a structured course context, the learners are assessed altogether by the same assignment or test. Hence we had to make the design choices focusing largely on individual residents rather than on their comparative performance analysis. We thus recommend designers to gain a better understanding of the training context, which would help them verify whether to adopt an existing visualization framework or to develop a new one.

The old system that preceded our dashboard was a tabular report file that had been in use for quite a while and so in designing the layout of our new dashboard we followed a similar tabular design but enriched it with visualizations to ensure users develop a sense of familiarity with the new dashboard. Further, to tackle the problem of information overload in this tabular design we grouped visual charts into collapsible blocks with headers to create an overview that can be expanded on demand. We visualize the essential information in a simple chart first but provide the ability for more complex investigations through advanced filters that can be brought up on demand.

Although we attempted to have a minimalist design, people needed some time to get used to the system, e.g., sometimes they did not notice how various filters could be used, and after pointing those out they mentioned that *"that is amazing!"* Through our iterative development, we observed that starting with a simple and intuitive design, and then adding new features incrementally, is often useful to help users to get acclimatized to the system and transition from novice to expert users.

Since the resident view was accessed and used by both residents and educators, the usage varied widely among the users. While our initial system design was developed for a desktop screen aspect ratio to be used in an office setting during competence meetings, we noticed through our user logs that a significant share of residents were accessing the dashboard on mobile devices before their hospital rotations to keep track of their progress. This meant that they were unable to access the entire feature set of the system as most advanced filters and charts looked distorted in a portrait screen mode

on mobile screens compared to the traditional landscape ratio of desktop monitors and laptop screens. In light of this, we adopted our design to be responsive to screen sizes and also created portrait variants of our charts that are shown in these alternate screen sizes.

## 8 LIMITATIONS AND FUTURE WORK

A natural limitation of our work comes from our exclusive focus on the emergency medicine department. However, we believe our findings will be applicable to visualize training data for various other departments. Recently, our visual platform has on-boarded 5 new programs (Internal Medicine, Pathology, Neurosurgery, Internal Medicine and Obstetrics/Gynecology) and several other programs within and beyond the university have also shown interest in adopting our system. We have also developed a similarly structured visualization system for the Undergraduate Medicine Program in our University and are currently studying its viability. One possible avenue for future research is to examine the generalizability of our visualization approach for other professional training scenarios.

The focus group participants were all from North America, and their feedback and opinions may differ from the medical professionals from other parts of the world. Whether the differences in the cultural aspects of how residents should be trained require different design considerations, would be an interesting direction to explore.

We only had a small number of reviewers (6) and residents (16) available. It would be interesting to examine the usage pattern with a larger number of users when other medical departments across Canada start using our platform. A significant byproduct of aggregating multiple resident records is that information can be now mined about the overall health of the learning program itself. A future design extension of our system could include looking at this information such as how residents acquire EPAs across different rotations or how EPA acquisition rates vary each month across the different programs. This could be valuable to program evaluators trying to improve the learning program and the quality of feedback given to residents.

## 9 CONCLUSION

In this paper we presented a visualization platform to assist various user groups (residents, reviewers, and program evaluators) to analyze residency training data in academic medicine. We conducted an uncontrolled longitudinal study over four months. Our analysis of the user logs and the change in learning metrics after the introduction of the system and feedback from various user groups showed that the developed visualizations were effectively used by the users in various learning and training processes. We believe that our study in this paper will inspire further visualization research in learning management systems of professional training scenarios.

### ACKNOWLEDGMENTS

The authors wish to thank Rob Woods, Robert Carey, Lynsey Martin and Teresa Chan for their support in this research. This project was sponsored by the College of Medicine Strategic Grant from the Office of the Vice Dean of Research at the University of Saskatchewan. The research of Debajyoti Mondal is supported in part by the Natural Sciences and Engineering Research Council of Canada (NSERC), and by a Canada First Research Excellence Fund (CFREF) grant.

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
