# OpenReview forum: "Scope and Impact of Visualization in  Training  Professionals in Academic Medicine"
_graphicsinterface.org/Graphics_Interface/2020/Conference — GI 2020_

### Official Review · AnonReviewer1 · 2020-02-07

**Rating:** 7
**Confidence:** 4

**Review:**

This paper presents a design study of a visualization system for tracking and reviewing resident physicians’ performance in a medical training program. The design of the system was through four focus groups and the evaluation was through a four-month deployment study. User behavior changes were observed after introducing the system.

Overall, I think this is a great paper. I appreciate the authors’ effort in designing and deploying a visualization system in real-world scenarios. The topic is definitely critical, because effective training programs are essential in providing high-quality medical care. Although there is not much novelty in the visualization design, I think this paper focuses on the process, insights, and patterns learned during the design and evaluation process. However, this paper can be stronger if there is more meat in the requirement analysis and design implication.

For the requirement analysis that was achieved through focus groups, this paper lists five questions to solve (four in the resident view, and one in the reviewer view). I’d like to see more regarding how these questions are derived. For example, this can be benefited from a summary table of the data that the focus groups want to visualize, a few figures of the sketches, and quotes of the users in the focus groups. Currently, it is still unclear whether these questions cover all the needs or are cherry picked. The design implication section lacks depth. The insights gained are mainly on usability, which in my view are not very insightful. I wish the authors can distill a set of higher level principles to inform the design of similar systems in the future.

Moreover, I think the reviewer view is too simplistic. The system allows the reviewer to compare residents in different time period, but does not have the ability to show “why” certain residents are underperformed. As a good system to support educational training, it should help the reviewer/educator identify problems of the weaker learners and help them. In other words, the comparison should inform actions. I’d like to see some discussion on this aspect in the paper.

---

### Official Review · AnonReviewer3 · 2020-02-12
**Review of Scope and Impact of Visualization in Training Professionals in Academic Medicine**

**Rating:** 7
**Confidence:** 3

**Review:**

This submission reports on the creation of a system to help medical residents and their reviewers to assess their learning using an information visualization dashboard, designed for and with them in a participatory process, deployed in their setting, and evaluated with them through a longitudinal study.

Quality
The methodology employed for conducting this research sources methods from diverse fields and is relevant.

Clarity
The presentation is very clear, with pertinent textual and visual explanations.

Originality
The review of related work is varied across relative disciplines and well positioned.

Signifiance
The system has been designed and developed and evaluated so that it ended up being useful to domain experts (medical residents and their reviewers).

I advocate for accepting this submission.

ABSTRACT

Abstract provides information that is ideally expected: one sentence of context, summary of contribution, explanation of system and methodology. I would suggest to use active voice instead of passive to clarify who contributed what ("The system was developed", "...was installed").

INTRODUCTION

The motivation and context is sound, with references on how information visualization and dashboards support learning analytics or educational data visualization.

The proposed methodology of design and development relies on well established practices: eliciting requirements through focus groups, designing using action design research framework, implementation through agile development, evaluating the system through uncontrolled longitudinal studies and feedback sessions. Obtained results are supported with clear metrics.

RELATED WORK

The related work is well balanced with a review on visualization dashboards and visualization in medical training with references from diverse related research communities.

"One reason for this gap seems to be the lack of collaboration among the developers, end-users and visualization experts."
The passive voice of the sentence does not help to identify who posited this reason: the authors of the submission or Vieira et al. [36]? Also, before initiating collaborations, I would say that all parties must first be aware of each others contributions, so I would rephrase the reason as a "lack of communication" among them.

APPLICATION BACKGROUND

This section conveniently introduces domain-specific terms and thus contributes to make the paper standalone in understanding the context. Requirement analysis was conducted through focus groups including active participation of domain experts (including involving them in sketching their desired features for data presentation). Data characterization is assorted with visibly clear understanding and explanation of the domain.

Q1 can be reformulated with plural to avoid gender bias (so that this is harmonized with similar efforts along the paper).

VISUALIZATION DESIGN

The rationale for visualization design is clearly explained and illustrated.

The choice for visualizing rotation schedules using an interval chart rather than a more space-consuming Gantt chart widespread in time/project management is smart.

The decisions on color scales adjustments to highlight under-performance while shadowing over-performance on EPA count per rotation is well motivated by contextual needs.

Figure 4: I would suggest to split the figure into 2 rows (3.5 and 3.6) and annotate columns in black font over white paper background, instead of white font over blue application background: with a low zoom level on my PDF reader, I had first confused these annotations with potential widgets in the application.

For further inspiration on visualization for comparing (resident) profiles, I'd suggest to browse other works by Plaisant et  al. in addition to [29]:
https://hcil.umd.edu/eventaction/
https://hcil.umd.edu/peerfinder/

IMPLEMENTATION DETAILS

The implementation details report on constraints that may be too project-specific (with occurrences of "project" or "the University") and would gain to be generalized. Congratulations for opensourcing the code to potentially help other institutions with medical programs ("across Canada", or beyond?).

The responsive design choice is great for multiple device access with various form factors. Rendering in SVG with d3 might pose issues regarding accessibility, where efforts for compliance are left at the discretion of application developers rather than library developers. See https://d3plus.org/accessibility/

USER EVALUATION AND FEEDBACK

The user evaluation and feedback proposes analysis of user logs that informed changes in metrics for measuring improvement in learning program once their system was adopted by residents and reviewers; and their feedback.

I would suggest the following references to inform analysis of user logs:
- H. Guo, S. R. Gomez, C. Ziemkiewicz and D. H. Laidlaw, "A Case Study Using Visualization Interaction Logs and Insight Metrics to Understand How Analysts Arrive at Insights," in IEEE Transactions on Visualization and Computer Graphics, vol. 22, no. 1, pp. 51-60, 31 Jan. 2016.
doi: 10.1109/TVCG.2015.2467613
- Papers from the IEEE VIS'16 Workshop: Logging Interactive Visualizations & Visualizing Interaction Logs
http://livvil.github.io/workshop/

DESIGN CHOICES AND INSIGHTS GAINED

I found the design considerations to be mostly obvious and known to designers and developers of user interfaces and information visualization.

LIMITATIONS AND FUTURE WORK

The limitations are mainly focused on the specificity of project requirements to one University in Canada, the small sample size of participants to evaluations.

SUPPLEMENTARY VIDEO

The video introduces the application domain and showcases diverse tasks supported by the tool presented in the submission.
Audio quality of the voice over could be improved with a proper microphone and recording settings.

---

### Official Review · AnonReviewer2 · 2020-02-12
**Good first step toward building visualization systems to support professional medical training**

**Rating:** 8
**Confidence:** 4

**Review:**

The authors have built a platform for assisting medical residents and their supervising doctors in keeping track of their progress toward different milestones. The system was designed over multiple iterations with feedback from the residents and supervisors and was finally deployed and monitored for a period of four months to collect in-the-wild usage data. The goal here was to understand the scope of visualizations to improve the training and learning process in a residency program.

The manuscript communicates the overall design process and several design decisions in-depth providing the readers with sufficient information to appreciate the efficiency and simplicity of the end-product. The authors have done a good job explaining the context and the relevant terminology. The requirement analysis, research questions, and resident/supervisor priorities are well explained and flow well in to the final design of the system. Finally, building such a system, completing the data pipeline, and maintaining it for over four months is considerable effort. Kudos in taking it beyond requirement gathering and building the system.

That said, there are some areas which can be improved upon:
- Although the authors did evaluate the performance of their system against historical data, there were a few things which I would have liked to see addressed in that comparison. There was no mention of the previous system or set-up which was used to review residents before the new system was implemented. Even if it was just manually parsing logs, that information would be vital in appreciating what was developed even more. Another thing that stood out in quantitative evaluation was the comparison of metrics from current deployment compared against those of the prior year. Given it was the same cohort, there would be some amount of learning or improvement which would have happened naturally over their first year and would have influenced their performance in the second year when the system was deployed. Some discussion around this would be useful to better appreciate the system.

- The design choices section on pg. 11 seems a bit superficial to be considered as generalized take-aways for future research. Maybe, instead of just focusing on what design choices worked, dedicating some space to discuss failure points would also be beneficial. What are some potential flaws or issues which future researchers should be aware of when building similar systems?

Overall, I think this is a good first step towards building visualization systems to aid in medical professional training.

---

### Meta-Review · Area_Chair1 · 2020-02-14

**Recommendation:** Accept
**Confidence:** 4

**Metareview:**

The reviewers are in agreement that this is a well-motivated paper and should be accepted. As R1 mentioned, the contribution does not lie in a novel visualization but rather in the process, insights, and patterns learned during the design and evaluation process.

The reviewers also agreed that the design implications section lacked depth. This is the one area where the paper has the biggest scope of improving. The reviewers have offered suggestions for different approaches to addressing this shortcoming.

Some other comments worth highlighting:
R1 has raised some concerns regarding how the 5 questions were selection and would like added details regarding the process.
R2 would like some discussion around the prior approach or set-up that this system replaced.
R3 has provided detailed feedback on minor changes which will improve the overall readability of the paper and can be accomplished prior to the camera ready submission.

---

### Decision · Program_Chairs · 2020-02-18

Accept